# Phenolic Profiling and Bioactive Properties of *Arthrospira platensis* Extract in Alleviating Acute and Sub-Chronic Colitis

**DOI:** 10.3390/ijms26125692

**Published:** 2025-06-13

**Authors:** Meriem Aziez, Ramona Suharoschi, Mohamed Sofiane Merakeb, Oana Lelia Pop, Călina Ciont, Floricuța Ranga, Riad Ferhat, Safia Affenai, Dan C. Vodnar, Angela Cozma, Adriana Fodor, Elhadia Mansouri, Dalila Smati, Noureddine Bribi

**Affiliations:** 1Laboratory of Plant Biotechnology and Ethnobotany, Faculty of Nature and Life Sciences, University of Bejaia, Bejaia 06000, Algeria; meriem.aziez@univ-bejaia.dz (M.A.); mohamedsofiane.merakeb@univ-bejaia.dz (M.S.M.); riad.ferhat@univ-bejaia.dz (R.F.); safia.affenai@univ-bejaia.dz (S.A.); noureddine.bribi@univ-bejaia.dz (N.B.); 2Molecular Nutrition and Proteomics Lab, Bld. Life Science Institute, Department of Food Science, University of Agricultural Science and Veterinary Medicine, 3-5 Calea Mănăstur, 400372 Cluj-Napoca, Romania; 3Food Biotechnology and Molecular Gastronomy, Bld. Life Science Institute, Department of Food Science, University of Agricultural Science and Veterinary Medicine, Calea Mănăstur 3-5, 400372 Cluj-Napoca, Romania; florica.ranga@usamvcluj.ro (F.R.); dan.vodnar@usamvcluj.ro (D.C.V.); 4Clinical Center of Diabetes, Nutrition and Metabolic Diseases, Cluj-Napoca, Faculty of Medicine, “Iuliu-Hațieganu” University of Medicine and Pharmacy, 400012 Cluj-Napoca, Romania; angelacozma@yahoo.com; 5Internal Medicine Department, 4th Medical Clinic “Iuliu Haţieganu” University of Medicine and Pharmacy, 400012 Cluj-Napoca, Romania; adriana.fodor@umfcluj.ro; 6Laboratory of Toxicology, Faculty of Pharmacy of Algiers, Algiers 16000, Algeria; 7Pharmacy Department, Faculty of Medicine, Benyoucef Benkhadda University, Alger Centre 16000, Algeria; dalila_smati@yahoo.fr

**Keywords:** spirulina, HPLC-DAD-ESI-MS, phenolic compounds, DNBS-induced colitis

## Abstract

*Arthrospira platensis*, a filamentous photosynthetic cyanobacterium, is widely recognized for its high nutritional value, broad spectrum of bioactive compounds, and excellent safety profile, making it a promising natural source for health-promoting applications. This study aimed to profile the phenolic constituents of an ethanolic extract of *A. platensis* (EAP) using HPLC-DAD-ESI-MS and to investigate its pharmacological effects in attenuating acute and sub-chronic experimental colitis, as well as its antioxidant and antifungal properties. Colitis was induced in BALB/c mice by intrarectal administration of 2,4-dinitrobenzenesulfonic acid (DNBS), followed by oral administration of EAP at doses of 50, 100, and 200 mg/kg. Phenolic profiling revealed eight major compounds, with a cumulative content of 6.777 mg/g of extract, with Pyrogallol, Ferulic acid, and Chlorogenic acid being the most abundant. In vivo, EAP treatment significantly reduced the Disease Activity Index (DAI), alleviated macroscopic colonic damage, and preserved colonic mucosal integrity in both inflammatory phases. Biochemical analyses revealed significant reductions in myeloperoxidase (MPO) activity, nitric oxide (NO), and malondialdehyde (MDA) levels, accompanied by increased reduced glutathione (GSH) content and catalase activity. In vitro, EAP demonstrated notable antioxidant effects, including 56% DPPH and 47% ABTS radical scavenging activities, and an 81% ferrous ion-chelating capacity. Furthermore, it exhibited antifungal activity, with inhibition zones of 20 mm against *Candida albicans* and 15 mm against *Aspergillus flavus*, respectively. These findings highlight the multitarget bioactivity of EAP and support its potential as a natural agent for managing intestinal inflammation and oxidative stress across both acute and sub-chronic phases.

## 1. Introduction

Crohn’s disease (CD) and ulcerative colitis (UC) are the two most prevalent forms of inflammatory bowel disease (IBD), characterized by chronic inflammation of the gastrointestinal (GI) tract [1,2]. Clinically, CD and UC present with similar symptoms, including abdominal pain, diarrhea, and weight loss, typically marked by alternating periods of remission and exacerbation [3,4,5,6]. Despite advances in understanding IBD, the prevalence of these diseases continues to rise globally, placing a significant burden on healthcare systems and patients’ quality of life [7,8,9]. Epidemiological studies have revealed significant geographic variability, with the highest incidence rates reported in industrialized countries and increasing rate in newly industrialized regions [7,10]. The exact cause of IBD remains unknown; however, its onset and progression are thought to result from a combination of environmental, microbial, and immunological factors in genetically predisposed individuals [11,12]. The main pathological feature of IBD is inflammation, which is strongly associated with the overproduction of reactive metabolites, such as reactive oxygen species (ROS) and reactive nitrogen species (RNS). These molecules can affect cellular macromolecules and induce imbalances in membrane integrity, leading to tissue damage in the gastrointestinal tract [13]. Current therapies aim to reduce inflammation using synthetic anti-inflammatory agents, corticosteroids, and immunosuppressants; however, their effectiveness is often limited by adverse side effects and the risk of long-term complications [14,15]. Consequently, there is growing interest in exploring active compounds from natural sources that may offer enhanced therapeutic efficacy and more favorable toxicological profiles [16,17,18].

*Arthrospira platensis*, commonly known as Spirulina, is a filamentous cyanobacterium of the Oscillatoriaceae family that has garnered significant attention for its nutritional and pharmacological potential [19,20,21]. Historically, *A. platensis* was harvested by the Kanembu people of Chad from the alkaline waters of Lake Chad and processed into sun-dried cakes known as *dihe*. These cakes were traditionally consumed as a nutrient-dense food source and were particularly valued for their ability to sustain energy during periods of food scarcity [22,23]. Similarly, the Aztecs of Central Mexico collected it from Lake Texcoco as a valuable protein source. It was incorporated into their diet and was also believed to enhance physical endurance and support general well-being [24]. These traditional practices, based on knowledge passed down through generations, anticipated modern scientific findings highlighting *A. platensis* as a rich source of proteins, essential fatty acids, vitamins, and minerals. Today, it is widely used as a dietary supplement and is classified as “Generally Recognized as Safe” (GRAS) by the U.S. Food and Drug Administration (FDA) [24,25,26,27,28]. Beyond its nutritional attributes, *A. platensis* is a potent source of bioactive compounds, including phycobiliproteins, polysaccharides, phenolic acids, and sterols, that exhibit diverse pharmacological activities, such as antioxidant, anti-inflammatory, hepatoprotective, and antiulcer effects [29,30,31,32,33,34,35]. Collectively, these bioactivities underscore the potential of Spirulina as a promising candidate for developing innovative therapeutic strategies, particularly in the context of IBD.

To the best of our knowledge, while previous studies have explored the therapeutic effects of whole Spirulina or its extracts in models of acute colitis (including those induced by dextran sulfate sodium (DSS) or acetic acid) [36,37,38], no studies have yet examined its effects at different stages of DNBS-induced colitis. The present study aimed to address this gap by employing a more detailed and integrative approach. It focused on the comprehensive phenolic profiling of an ethanolic extract of *Arthrospira platensis* (EAP) using High-Performance Liquid Chromatography with Diode Array Detection coupled to Electrospray Ionization Mass Spectrometry (HPLC-DAD-ESI-MS) and evaluated its pharmacological potential throughout the different stages of colitis (from the acute to the sub-chronic phase) using the 2,4-dinitrobenzenesulfonic acid (DNBS)-induced model, which closely mimics human Crohn’s disease. In this study, sub-chronic colitis refers to a model of prolonged intestinal inflammation induced by repeated DNBS administration [39,40,41]. Additionally, in vitro assays were performed to elucidate the antioxidant and antifungal activities of EAP, thereby providing broader insights into its potential therapeutic applications in the context of IBD.

## 2. Results

### 2.1. Phenolic Profile of EAP Extract

The phenolic composition of the ethanolic extract of *A. platensis* was analyzed using the HPLC-DAD-ESI-MS, allowing for both qualitative and quantitative characterization. The chromatographic profile of the identified compounds is presented in Figure 1, while their respective concentrations and mass-to-charge ratios ([M + H]^+^) are summarized in Table 1. A total of eight phenolic compounds were identified in the ethanolic extract, with a cumulative content of 6.777 mg/g of dry extract. Pyrogallol was the predominant compound, representing 46.36% of the total identified phenolics, followed by ferulic acid (15.10%) and chlorogenic acid (10.37%). These findings indicate the chemical richness and diversity of phenolic constituents present in EAP.

Table 1 lists the identified phenolic compounds, their retention times (Rt), maximum absorption wavelengths (λmax), mass-to-charge ratios ([M + H]^+^), and concentrations (mg/g of dry extract). Compounds were quantified using calibration curves constructed from corresponding reference standards.

### 2.2. Effect of EAP on Acute Colitis

#### 2.2.1. Clinical and Morphological Parameters

The intestinal anti-inflammatory activity of orally administered EAP was investigated in the DNBS-induced acute colitis model. In BALB/c mice, DNBS administration resulted in a significant and progressive increase in the disease activity index (DAI) in the untreated colitic group compared with the healthy control group (*** *p* < 0.001), reflecting severe colonic inflammation. However, oral treatment with EAP (50, 100, and 200 mg/kg) for three consecutive days led to a significant reduction in the DAI compared to the untreated colitic group (Figure 2A).

Macroscopically, DNBS-induced acute colitis was associated with a notable thickening and shortening of the colon, as evidenced by a significantly increased colon weight-to-length ratio (47.11 ± 5.80) in the untreated colitic group compared with the healthy group (26.21 ± 1.34; *** *p* < 0.001). Treatment with EAP significantly reduced this ratio, with values of 37.20 ± 2.58, 36.73 ± 4.31, and 45.92 ± 0.96 for the 50, 100, and 200 mg/kg doses, respectively (Figure 2B,C).

#### 2.2.2. Histological Evaluation

The protective effects of EAP were further confirmed by histological analysis. In the non-colitic group, the colonic architecture was well-preserved, exhibiting intact epithelial structures and an absence of inflammatory infiltration (Figure 3A). However, the microscopical examination of colonic samples from the untreated colitic group revealed a submucosal edema, considerable infiltration of inflammatory cells in the submucosa, and focal deterioration of the mucosa (including disruption of crypt integrity and necrosis of epithelial cells), further underscoring the severity of DNBS-induced inflammation (Figure 3B).

Treatment with different doses of EAP resulted in a significant decrease in colonic lesions. Amelioration of the histological appearance of the mucosa and submucosa was observed mainly in the colonic samples from mice treated with dose of 100 mg/kg of EAP. The histological results obtained in the groups of mice treated with EAP extract allowed to highlight the intestinal anti-inflammatory effects of *A. platensis*. This anti-inflammatory effect was confirmed by a reduction in the infiltration of immune cells and the preservation of intact crypt structure (Figure 3C–E).

#### 2.2.3. Biochemical Markers

To further elucidate the protective effects of the ethanolic extract of *A. platensis*, we measured key biochemical markers of colonic inflammation and oxidative stress. Lipid peroxidation and oxidative tissue damage were evaluated by determining malondialdehyde (MDA) accumulation in colonic tissues. We observed a significant increase in MDA levels (10.91 ± 0.76 nM/mg of colon) in the untreated colitic group compared with the non-colitic group (5.85 ± 0.87 nM/mg of colon; **** p* < 0.001). In contrast, all doses of EAP (50, 100 and 200 mg/kg) significantly reduced the MDA levels (to 8.17 ± 0.45, 8.14 ± 1.04 and 8.98 ± 1.36 nM/mg of colon, respectively) (Figure 4A).

Nitrite levels, indicative of nitric oxide (NO) production, were also significantly increased in the untreated colitic group (22.02 ± 0.98 µM/100 mg of colon) when compared with the control group (14.39 ± 1.79 µM/ 100 mg of colon; *** *p* < 0.001). EAP administration significantly attenuated this elevation, yielding nitrite levels of 17.50 ± 3.02, 15.39 ± 2.66, and 17.45 ± 1.45 µM/100 mg of colon at doses of 50, 100, and 200 mg/kg, respectively (Figure 4B).

Finally, we evaluated the GSH levels as an indicator of antioxidant defense. A significant decrease in GSH levels was observed in the untreated colitic group (3.60 ± 1.09 µM/100 mg of protein) compared with the control group (16.19 ± 2.94 µM/100 mg of proteins; *** *p* < 0.001). However, EAP treatment significantly restored GSH levels, with values of 6.29 ± 2.47, 8.02 ± 1.58, and 5.23 ± 0.91 µM/100 mg of protein for the 50, 100, and 200 mg/kg groups, respectively (Figure 4C).

### 2.3. Effect of EAP on Sub-Chronic Colitis

#### 2.3.1. Clinical and Morphological Parameters

The intestinal anti-inflammatory activity of orally administered EAP was investigated in the DNBS-induced sub-chronic colitis. The second DNBS administration to BALB/c mice induced a progressive increase in DAI values. Oral treatment of colitic mice with increasing doses (50, 100, and 200 mg/kg) of EAP progressively reduced the DAI compared with the untreated colitic group (Figure 5A).

In addition, colonic inflammation was also associated with an increase in the weight/length ratio of the colitic group (47.85 ± 3.70) in comparison with the healthy untreated mice (36.55 ± 3.44; *** *p* < 0.001). The untreated colitic mice showed a thickening and shortening of the colon, leading to an increase in this ratio. This is due to the fact that Crohn’s disease causes chronic inflammation and fibrosis, which gradually lead to a narrowing of the colon and significant structural changes in the intestinal tissue [42]. A significant reduction in the colonic weight/length ratio was observed following treatment with the ethanolic extract of *A. platensis*, with values of 42.69 ± 2.17, 42.63 ± 2.07, and 39.03 ± 3.90 for the doses of 50, 100, and 200 mg/kg, respectively, compared to the colitic group, confirming the attenuation of colitis severity (Figure 5B,C).

#### 2.3.2. Histological Evaluation

The beneficial effects of EAP were confirmed histologically. Colonic tissue of the non-colitic control group showed no histological abnormalities (Figure 6A). In contrast, microscopic analysis of colonic tissue from the untreated colitic group revealed severe chronic and transmural colitis, characterized by pronounced submucosal edema, extensive infiltration of inflammatory cells within the submucosa, and focal mucosal damage, including crypt disorganization and the presence of granulomas (Figure 6B). The presence of transmural damage and granulomas is a key feature of Crohn’s disease [43,44]. In clear contrast, treatment with EAP at 50, 100, and 200 mg/kg resulted in less submucosal edema and reduced inflammatory cell infiltration with no architectural damage. These improvements were most prominent in colonic samples from mice treated with the 200 mg/kg dose (Figure 6C–E).

#### 2.3.3. Biochemical Markers

The MPO activity of colon homogenates from all groups is presented in Figure 7A. MPO activity was significantly increased in the untreated colitic group (54.75 ± 8.73 mM/min/g of colon), compared with the non-colitic group (11.70 ± 3.69 mM/min/g of colon; *** *p* < 0.001). Previous studies have shown that MPO activity, an indicator of neutrophil infiltration, increases with the severity of inflammatory damage in the colitic group [45]. This increase was significantly reduced in colitic mice treated with EAP, with values of 38.81 ± 4.89, 29.25 ± 6.08, and 24.57 ± 2.94 mM/min/g of colon, respectively. A reduction in MPO activity can be interpreted as a manifestation of an anticolitic effect [46].

NO levels were found to increase in the untreated colitic group (10.14 ± 19.55 µM/100 mg of colon) as compared with the non-colitic group (3.26 ± 5.22 µM/100 mg of colon; *** *p* < 0.001). Excess NO is responsible for increased vasodilation, vascular permeability, and tissue lesions, and it contributes to intestinal inflammation [47]. Administration of EAP to colitic mice led to a significant reduction in nitrite levels, as shown in Figure 7B. All tested EAP doses (50, 100, and 200 mg/kg) resulted in decreased nitrite concentrations (5.78 ± 1.48, 5.51 ± 1.31, and 3.63 ± 1.03 µM/100 mg of colon, respectively) compared with the untreated colitic control group. MDA content in colonic tissues was used as an indicator of lipid peroxidation and cellular damage. We observed a significant increase in MDA levels in the untreated colitic group (8.94 ± 1.18 µM/g of colon) compared with the non-colitic group (3.04 ± 1.26 µM/g of colon; *** *p* < 0.001). This increase was beneficially modulated when animals were treated with different doses (50, 100 and 200 mg/kg) of EAP (resulting in levels of 5.33 ± 1.23, 3.87 ± 0.08, and 4.85 ± 0.62 µM/g of colon, respectively) (Figure 7C).

DNBS instillation led to a significant decrease in Catalase activity (52.00 ± 16.57 mM/min/g of colon) and GSH levels (5.19 ± 1.54 µM/100 mg of protein), respectively, compared with the non-colitic group (catalase activity: 156.21 ± 23.81 mM/min/g of colon. GSH: 18.16 ± 2.06 µM/100 mg of proteins) (*** *p* < 0.001 for both comparisons). EAP counteracted the deleterious effect of DNBS by significantly increasing GSH levels (to 13.31 ± 1.39, 12.21 ± 1.28, and 9.11 ± 2.60 µM/100 mg of protein, for the doses of 50, 100, and 200 mg/kg, respectively). However, the increase in catalase activity was not significant (*p* > 0.05) (Figure 7D,E). Catalase and GSH can scavenge free radicals and ROS, thereby preventing oxidative stress-induced lipid peroxidation and cell damage [48].

### 2.4. Antioxidant Activities

To evaluate the antioxidant capacity of EAP, 2,2-diphenyl-1-picrylhydrazyl (DPPH) and 2,2′-azino-bis (3-ethylbenzothiazoline-6-sulfonic acid) (ABTS) radical-scavenging assays along with the ferrous ion-chelating (FIC) assay, were performed. In all test models, EAP exhibited a strong antioxidant capacity, with activity progressively increasing in a concentration-dependent manner. At a concentration of 1 mg/mL, the scavenging activities for DPPH• and ABTS•^+^ were 56.14 ± 0.5% and 47.16 ± 0.07%, respectively, and the FIC activity was 81.21 ± 1.25%. These were compared to Trolox (DPPH•: 93.39 ± 0.20%. ABTS•⁺: 98.99 ± 0.067%. FIC: 95.51 ± 0.12%) and ascorbic acid (DPPH•: 97.91 ± 0.13%. ABTS•⁺: 98.85 ± 0.067%. FIC: 97.06 ± 0.18%) (Figure 8A–C).

Radical scavenging activity is indirectly related to the IC_50_ value. According to Table 2, the IC_50_ values for EAP were 0.871 ± 0.012 mg/mL (DPPH•), 0.675 ± 0.026 mg/mL (ABTS•⁺), and 0.213 ± 0.001 mg/mL (FIC). These were compared with Trolox (IC_50_: 0.081 ± 0.001 mg/mL for DPPH•, 0.068 ± 0.00009 mg/mL for ABTS•^+^, and 0.081 ± 0.001 mg/mL for FIC) and ascorbic acid (IC_50_: 0.067 ± 0.00004 mg/mL for DPPH•, 0.072 ± 0.0001 mg/mL for ABTS•⁺, and 0.083 ± 0.00 mg/mL for FIC).

### 2.5. Antifungal Activity

The antifungal activity of EAP was evaluated against *Candida albicans* and *Aspergillus flavus*, with results summarized in Table 3. EAP exhibited a concentration-dependent inhibitory effect where the 100 mg/mL dose showed high antifungal activity against the two microorganisms, with inhibition zone diameters of 20 ± 0.1 mm for *C. albicans* and 15 ± 0.1 mm for *A. flavus*. In comparison, the reference antifungal agent ketoconazole (10 µg/disc) demonstrated superior activity, producing inhibition zones of 30 ± 0.05 mm for *C. albicans* and 20 ± 0.05 mm for *A. flavus*.

## 3. Discussion

This study explored the therapeutic potential of EAP with a particular emphasis on intestinal inflammation and oxidative stress. By combining comprehensive phenolic profiling with both in vivo and in vitro assessments, this study provides valuable mechanistic insights into the EAP’s multifunctional bioactivity and its potential relevance in managing chronic intestinal disorders.

HPLC-DAD-ESI-MS analysis confirmed the presence of eight phenolic compounds in EAP. Pyrogallol, ferulic acid, and chlorogenic acid were identified as major constituents, each known for potent antioxidant and anti-inflammatory activities, underscoring EAP’s pharmacological potential in chronic inflammatory conditions [49,50,51]. Compared with previous reports, EAP exhibited a higher total phenolic content but reduced chemical diversity. Compared with the 24 phenolic compounds (with concentrations ranging from 0.73 to 1.24 mg/g) identified by Uzlasir et al. [52], our EAP shared several key compounds, including ferulic acid, chlorogenic acid, p-coumaric acid, vanillic acid, and 3-hydroxybenzoic acid. The lower diversity of compounds observed in our study may reflect methodological and biological differences, such as strain specificity, geographical origin, and cultivation conditions, all of which can influence the metabolic composition of *Spirulina* [53,54,55].

The therapeutic potential of EAP against colonic inflammation was investigated using both acute and sub-chronic DNBS-induced colitis models. Doses of 50, 100, and 200 mg/kg were selected based on previously reported toxicological evaluations demonstrating the safety of Spirulina extracts at these concentrations and are consistent with published preclinical studies on colitis models using Spirulina [19,36,37]. This dosing range was strategically designed to encompass low to moderate exposures, allowing for the investigation of dose-dependent therapeutic responses while maintaining a favorable safety profile. Each experimental group comprised seven animals (n = 7), a sample size widely accepted in experimental colitis models for providing sufficient statistical power to detect significant biological differences while remaining in compliance with ethical standards for animal research. In the acute model, a short-term (3-day) oral treatment with EAP significantly attenuated the early signs of colonic inflammation. These effects were highlighted by a significant reduction in the DAI, improvements in the macroscopic appearance of the colon, and the normalization of the colon weight-to-length ratio. Biochemically, EAP decreased MDA and NO levels and preserved GSH levels, highlighting its early antioxidant effect. These observations suggest that EAP can rapidly counteract oxidative stress and modulate early inflammatory responses by preventing the amplification of tissue injury at the onset of the disease.

In the sub-chronic model, which mimics many manifestations of human Crohn′s disease, the inflammatory process becomes more complex and persistent [43,44]. This experimental model is associated with an imbalance comprising increased ROS and decreased antioxidant activity [56]. Administration of DNBS motivates the migration of polymorphonuclear neutrophils, an important source of ROS and RNS, which result in the release of MPO [57]. In addition, intestinal oxidative damage promotes lipid peroxidation and significantly impairs the antioxidant defense system, thereby contributing to mucosal injury and inflammation within the intestinal tract [48]. Treatment with EAP reduced local inflammation by decreasing MPO and NO production while promoting tissue healing through a reduction in MDA and enhancement of the antioxidant defense system. Although EAP treatment led to an increase in catalase activity in the sub-chronic colitis model, this effect did not reach statistical significance. This may reflect a selective enhancement of glutathione-related antioxidant responses, suggesting that catalase activity is less sensitive to modulation by EAP.

In the acute phase, the 100 mg/kg dose of EAP was found to be the most effective across several key parameters. This suggests that in the early stages of colitis, this dose provides an optimal balance between efficacy and safety. In contrast, in the sub-chronic phase, the 200 mg/kg dose proved to be more effective, highlighting the dose-dependent nature of the therapeutic response. The higher dose in the sub-chronic model was likely necessary to address the more persistent and complex inflammatory processes associated with prolonged disease.

The biochemical markers analyzed in this study (MPO and catalase activities, as well as GSH, MDA, and NO levels) provide critical insights into the underlying molecular mechanisms of EAP’s therapeutic effects. Specifically, the reduction in MPO activity points to a potential inhibition of the nuclear factor kappa B (NF-κB) pathway, which governs the expression of key pro-inflammatory cytokines, such as tumor necrosis factor-alpha (TNF-α) and interleukin-6 (IL-6) [58]. Simultaneously, the observed increase in GSH and catalase activity, along with the decrease in MDA, suggests the activation of the nuclear factor erythroid 2-related factor 2/antioxidant response element (Nrf2/ARE) signaling pathway, which orchestrates antioxidant responses and protects against oxidative damage [59,60]. Additionally, modulation of NO levels implicates a downregulation of the inducible nitric oxide synthase (iNOS) pathway, thereby mitigating nitrosative stress and preventing exacerbation of the inflammatory response [61].

Previous studies have reported the anti-inflammatory and antioxidant effects of *A. platensis* in other experimental acute colitis models [36,37,38,62]. While other bioactive compounds in *A. platensis*, such as phycocyanin, polysaccharides, and proteins, may contribute to these therapeutic effects, the effects observed in our study are likely mediated largely by its phenolic constituents through distinct molecular mechanisms. Pyrogallol has been shown to inhibit the NF-κB pathway, which reduces the expression of pro-inflammatory cytokines, such as TNF-α and IL-6 [63]. Chlorogenic acid and ferulic acid modulate the cyclooxygenase-2 (COX-2) pathway and activate Nrf2/ARE signaling, which enhances antioxidant defenses and suppresses lipid peroxidation [50,51,64]. In addition, these compounds regulate the iNOS pathway, limiting the production of reactive nitrogen species and mitigating nitrosative stress, which is crucial for reducing inflammation [65,66]. Taken together, these findings suggest that the therapeutic effects of EAP are largely attributable to its phenolic compounds, which target multiple signaling pathways involved in oxidative stress and inflammation, thereby contributing to its efficacy in treating colitis.

Complementary in vitro assays further confirmed the antioxidant capacity of the EAP extract, including DPPH and ABTS radical scavenging activities, as well as the ferrous ion-chelating activity, all of which were conducted in triplicate to ensure reproducibility and minimize experimental variability. This effect is likely mediated by the presence of phenolic compounds capable of scavenging free radicals and contributing to redox homeostasis [67,68]. Given the important role of oxidative stress in developing inflammatory and degenerative diseases, these properties enhance the therapeutic relevance of EAP [69,70]. While the chemical-based methods employed offer reliable and standardized insights into the radical-scavenging and metal-chelating capacities of EAP, we acknowledge that complementary cellular antioxidant assays (such as intracellular ROS quantification or the assessment of antioxidant enzyme induction) would provide a more mechanistic and biologically relevant understanding. Future investigations will, therefore, aim to incorporate such cell-based assays, particularly in intestinal epithelial models, to further elucidate the EAP’s antioxidant potential.

Additionally, the extract displayed potent antifungal activity against *Candida albicans* and *Aspergillus flavus*, consistent with previous findings [71,72,73]. This analysis was prompted by evidence implicating fungal dysbiosis, particularly *Candida albicans* overgrowth, in the exacerbation of intestinal inflammation in Crohn’s disease [74]. Given that the DNBS model mimics key characteristics of Crohn’s disease, assessing the antifungal activity *of* EAP may provide insight into its potential role in supporting microbial balance and intestinal barrier integrity. The inclusion of *A. flavus* broadened the antifungal screening to evaluate the EAP’s wider spectrum of activity. Although EAP produced inhibition zones ranging from 15 to 20 mm (at 100 mg/mL), ketoconazole (10 µg/disc) showed superior activity with inhibition zones of 30 mm and 20 mm against *C. albicans* and *A. flavus*, respectively. This indicates that EAP has moderate antifungal efficacy, suggesting potential as an adjunct or complementary therapy rather than a standalone treatment. However, the absence of Minimum Inhibitory Concentration (MIC) determination limits the comprehensive evaluation of its antifungal potential, indicating that further studies are needed to establish these values and better define its antifungal efficacy. This dual antioxidant–antifungal profile highlights the broad-spectrum bioactivity of EAP and suggests its potential as a complementary therapeutic agent against chronic intestinal inflammation associated with oxidative stress.

## 4. Materials and Methods

### 4.1. Biological Material and Reagents

The dry biomass of *A. platensis* was obtained in October 2021 from Kasdi Merbah University of Ouargla, Algeria. Reagents and chemicals were primarily sourced from Sigma-Aldrich (Madrid, Spain), except for Sabouraud Dextrose Agar (SDA), which was purchased from HiMedia (Mumbai, India). HPLC-grade acetonitrile was supplied by Merck (Darmstadt, Germany), and ultrapure water was produced using a Direct-Q UV purification system (Millipore, Burlington, MA, USA). Analytical standards with ≥99% purity (HPLC grade) were obtained from Sigma (St. Louis, MO, USA).

### 4.2. Ethanolic Extraction Process

The dried biomass of *A. platensis* was macerated in 80% ethanol at a 1:10 (*w*/*v*) ratio under continuous stirring for 24 h at 25 °C in the dark. The extract was then filtered using Whatman No. 1 filter paper and concentrated by drying at 40 °C, following the method described by Sb (2018) [75].

### 4.3. Phenolic Profiling of EAP Extract Using HPLC-DAD-ESI-MS

Phenolic compounds in the ethanolic extract of *A. platensis* were identified and quantified using high-performance liquid chromatography with diode-array detection and electrospray ionization mass spectrometry (HPLC-DAD-ESI-MS), as described by Călinoiu and Vodnar [76]. Analyses were performed on an HP-1200 system (Agilent Technologies, Santa Clara, CA, USA) equipped with a quaternary pump, autosampler, DAD detector, and an MS-6110 single-quadrupole API-ESI detector. Detection was conducted in positive ionization mode.

Chromatographic separation was achieved using a Kinetex XB-C18 column (5 µm, 4.5 × 150 mm; Phenomenex, Torrance, CA, USA). The mobile phase consisted of solvent A (0.1% formic acid in water) and solvent B (0.1% formic acid in acetonitrile), applied using a multistep gradient: 5% B for 2 min, ramped to 90% B over the next 20 min, held at 90% B for 4 min, then returned to 5% B over 6 min. The total run time was 30 min, with a flow rate of 0.5 mL/min and the column temperature was maintained at 25 ± 0.5 °C.

Mass spectrometric detection was performed in Scan mode under the following conditions: gas temperature 350 °C, nitrogen flow 7 L/min, nebulizer pressure 35 psi, capillary voltage 3000 V, fragmentor voltage 100 V; and mass range m/z 120–1500. Chromatograms were recorded at λ = 280 nm and λ = 340 nm. Data were acquired using Agilent ChemStation software, Rev. B.02.01-SR2. The reported phenolic composition values correspond to a single measurement obtained from one analytical run, without replicates.

Despite any instrumental limitations, the use of HPLC coupled with DAD and ESI-MS provided a robust and reliable platform for the characterization of phenolic compounds within the EAP. The method employed in this research was internally validated and standardized in our laboratory, allowing for the simultaneous collection of chromatographic, UV–Visible, and mass spectral data, which enabled effective compound profiling and tentative identification. The DAD facilitated the detection of characteristic UV absorbance maxima, supporting the classification of phenolic subclasses, while the ESI-MS provided key molecular ion information. The combination of these detection techniques enhanced confidence in compound identification and enabled comprehensive profiling, even in complex matrices.

### 4.4. Animals and Grouping

Seventy female BALB/c mice (20–25 g body weight) were obtained from the Pasteur Institute (Algiers, Algeria) and housed under standard conditions (25 ± 2 °C, 12 h light/dark cycle) with ad libitum access to food and water. All procedures complied with Directive 2010/63/EU and were approved by the Local Ethics Committee of the Laboratory of LBVE (Ref. No. CE-LBVE-2022-112).

Mice were randomly assigned to two experimental colitis models (acute and sub-chronic), each comprising five groups (n = 7 per group). In both models, Group I served as the non-colitic control group. There were also four DNBS-induced colitis groups: Group II was the untreated colitic group, while Groups III, IV, and V received oral administration of EAP at doses of 50, 100, and 200 mg/kg, respectively.

#### 4.4.1. Induction of Acute Colitis

Acute colitis was induced by a single intrarectal administration of DNBS, followed by three consecutive days of oral treatment with EAP, as described by Aziez et al. [42] (Figure 9A).

#### 4.4.2. Induction of Sub-Chronic Colitis

Sub-chronic colitis was induced using the method described by Martín et al. [41], with repeated DNBS administration to closely mimic the persistent inflammation characteristic of human Crohn’s disease. Briefly, colitis was induced by two intrarectal injections of DNBS spaced 12 days apart, and EAP was administered orally for seven consecutive days (from day 8 to day 14) (Figure 9B).

### 4.5. Colitis Severity Scoring and Histopathological Assessment

Clinical symptoms were monitored daily, starting from the first DNBS administration in the acute colitis model and from the second DNBS administration in the sub-chronic colitis model. The DAI was assessed based on body weight loss and other clinical parameters, including stool consistency, fecal bleeding, perianal wetness, piloerection, and reduced physical activity. At the end of each protocol, animals were sacrificed and dissected. The entire colon was removed, weighed, and its length was measured. To assess microscopic damage, colonic tissues were fixed in 4% buffered formaldehyde, embedded in paraffin, and sectioned at 5 μm using a rotary microtome. Sections were mounted on silane-coated glass slides, stained with hematoxylin and eosin (H&E), and examined under a Leica light microscope for histopathological assessment.

### 4.6. Evaluation of Biochemical Parameters

Colonic tissues were homogenized in 0.5% hexadecyl-trimethyl-ammonium bromide (HTAB) solution using a motor-driven homogenizer (HG-15A homogenizer, Daihan Scientific, Wonju, South Korea) [77]. After sequential centrifugation to remove debris, the post-mitochondrial supernatant (PMS) was collected for biochemical analysis [78].

#### 4.6.1. Myeloperoxidase Activity

MPO activity was measured spectrophotometrically by incubating the PMS with 50 mM phosphate buffer (pH 6.0), o-dianisidine hydrochloride (0.167 mg/mL), and 0.0005% hydrogen peroxide, as previously described [79]. The change in absorbance was recorded at 460 nm over 3 min. MPO activity was expressed as mM/min/g of tissue, using an extinction coefficient of 11.3 M^−1^ cm^−1^.

#### 4.6.2. Nitric Oxide Level

Nitrite levels, indicative of NO production, were quantified using the Griess reaction, as described by Aziez et al. [42]. Following protein precipitation with 10% trichloroacetic acid (TCA) and centrifugation, the supernatant was incubated with Griess reagents for 20 min in the dark. Absorbance was measured at 543 nm, and nitrite concentrations were calculated using a sodium nitrite (NaNO_2_) standard curve (ranging from 1 to 128 µM), with results expressed as µM/100 mg of tissue.

#### 4.6.3. Malondialdehyde Level

Lipid peroxidation in colonic tissue was evaluated by measuring MDA levels, following the method of Merakeb et al. [79]. Following protein precipitation with 35% TCA and centrifugation, the supernatant was mixed with sodium dodecyl sulfate (SDS), acetic acid, thiobarbituric acid (TBA), and distilled water. The mixture was heated to 95 °C, cooled to 4 °C, and the absorbance was recorded at 532 nm. MDA concentrations were calculated using an extinction coefficient of 1.56 × 10^5^ M^−1^·cm^−1^ and expressed as µM/g of tissue.

#### 4.6.4. Catalase Activity

Catalase activity in tissue supernatants was measured as described by Merakeb et al. [79]. The assay mixture contained 0.1 M phosphate buffer (pH 7.4), 0.019 M hydrogen peroxide, and the PMS. The decomposition of hydrogen peroxide was monitored by recording the decrease in absorbance at 240 nm at 30-s intervals over 3 min. Catalase activity was expressed as mM/min/g of tissue.

#### 4.6.5. GSH Level

GSH content in tissue supernatants was determined according to the method of Aziez et al. [42]. Following protein precipitation with 4% sulfosalicylic acid and centrifugation, the supernatant was mixed with 0.1 M phosphate buffer (pH 7.4) and 0.4% 5,5′-dithiobis-(2-nitrobenzoic acid) (DTNB). Absorbance was measured at 412 nm within 5 min. GSH concentrations were expressed as µM/100 mg of protein. Total protein content was quantified using the Bradford method with a bovine serum albumin (BSA) standard curve (ranging from 0 to 100 µg/mL).

### 4.7. In Vitro Antioxidant Activities

The antioxidant activity of the EAP extract was determined by DPPH and ABTS radical scavenging activities, as well as the FIC assay. All tests were performed in triplicate.

#### 4.7.1. DPPH Scavenging Assay

The DPPH• radical scavenging assay was performed according to the modified method of Blois et al. [80]. A reaction mixture containing 1 mL of DPPH^•^ solution (0.002% *w*/*v* in ethanol) and 0.5 mL of EAP (ranging from 0.1 to 1 mg/mL) was incubated in the dark at room temperature for 30 min. Absorbance was recorded at 517 nm. Trolox and ascorbic acid serve as reference antioxidants. The percentage of DPPH• scavenging activity was calculated, and the IC_50_ value (defined as the extract concentration required to inhibit 50% of DPPH• radicals) was determined.

#### 4.7.2. ABTS Scavenging Assay

The ABTS•^+^ radical scavenging assay was performed according to the modified method of Re et al. [81]. ABTS^•+^ radicals were generated by reacting 7 mM ABTS with 2.45 mM potassium persulfate and incubating the mixture in the dark for 16 h. The solution was then diluted with ethanol to achieve an absorbance of 0.7 at 734 nm. EAP (at concentrations ranging from 0.1 to 1 mg/mL) was added and incubated for 7 min at room temperature. Absorbance was measured at 734 nm. Trolox and ascorbic acid served as reference antioxidants. Antioxidant activity was expressed as the percentage of ABTS•⁺ scavenging, and IC_50_ values were calculated as the concentration of EAP required to reduce radical absorbance by 50%.

#### 4.7.3. Ferrous Ion-Chelating Assay

The FIC activity was assessed according to the modified method of Le et al. [82]. The reaction mixture consisted of EAP (at concentrations ranging from 0.1 to 1 mg/mL), 0.6 mM FeCl_2_, and methanol. After a 5-min incubation, ferrozine (5 mM) was added and the mixture was incubated for an additional 10 min to complex the residual Fe^2+^ ion. Absorbance was recorded at 562 nm. Trolox and ascorbic acid were used as reference standards. Chelating activity was expressed as a percentage of Fe^2+^ chelation, and IC_50_ values were calculated as the EAP concentration required to chelate 50% of ferrous ions.

### 4.8. Antifungal Activity

The antifungal activity of EAP was evaluated against *Candida albicans* (ATCC 10231) and *Aspergillus flavus* (ATCC 204304) using the disk diffusion method, as described by Collins et al. [83]. Fungal cultures were grown on SDA for 48 h to obtain young, actively growing cultures. Inocula were prepared to a turbidity equivalent to the 0.5 McFarland standard using sterile physiological water. Sterile disks impregnated with EAP (at concentrations of 50 and 100 mg/mL) were placed on inoculated agar plates. Plates inoculated with *C. albicans* were incubated at 32 °C for 24 h, and those with *A. flavus* were incubated at 25 °C for 5 days [84]. Ketoconazole, a synthetic antifungal antibiotic, was used as a positive control (10 µg/disc). All tests were performed in triplicate, and the diameters of the growth inhibition zones were measured.

### 4.9. Statistical Analysis

Data are expressed as the mean ± standard deviation (SD) from seven animals per group (in vivo studies) or three independent experiments (in vitro studies), as appropriate. Statistical analyses were performed using GraphPad Prism version 8.0.2.263 (GraphPad Software, San Diego, CA, USA). Group comparisons were conducted using one-way analysis of variance (ANOVA). Dunnett’s multiple comparisons test was applied for in vivo data, while Tukey’s post hoc test was used for in vitro antioxidant and antifungal assay data. A *p*-value < 0.05 was considered statistically significant.

## 5. Conclusions

This study highlights the therapeutic potential of the ethanolic extract of *A. platensis*, based on comprehensive chemical characterization and complementary in vivo and in vitro investigations. Enriched with bioactive phenolic compounds, EAP demonstrated anti-inflammatory, antioxidant, and antifungal properties by alleviating intestinal inflammation, protecting colonic architecture, and reducing oxidative and nitrosative stress during both acute and sub-chronic phases of experimental colitis. These findings emphasize the dual-phase efficacy of *the extract* and its potential as a natural agent for combating intestinal inflammation and oxidative stress. However, the absence of data on key inflammatory signaling pathways (such as NF-κB, COX-2, TNF-α, or IL-6) represents a limitation of the current study, as it prevents a full mechanistic understanding of the observed anti-inflammatory effects. These findings contribute to advancing the exploration of Spirulina-based natural strategies for managing inflammatory and oxidative disorders.

## Figures and Tables

**Figure 1 ijms-26-05692-f001:**
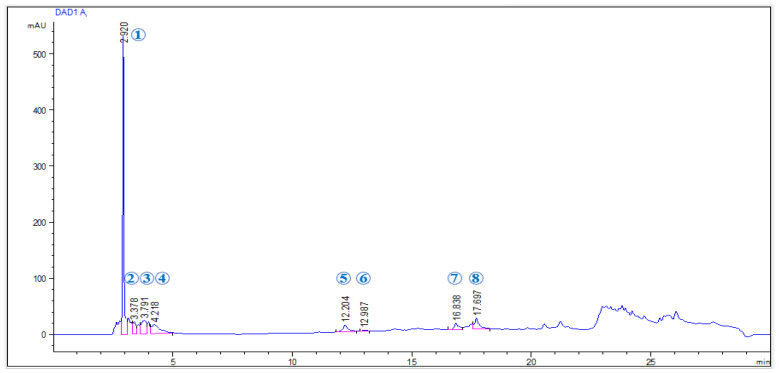
DAD chromatogram of phenolic compounds in the ethanolic extract of *A. platensis* (EAP) obtained by HPLC-DAD-ESI-MS analysis.

**Figure 2 ijms-26-05692-f002:**
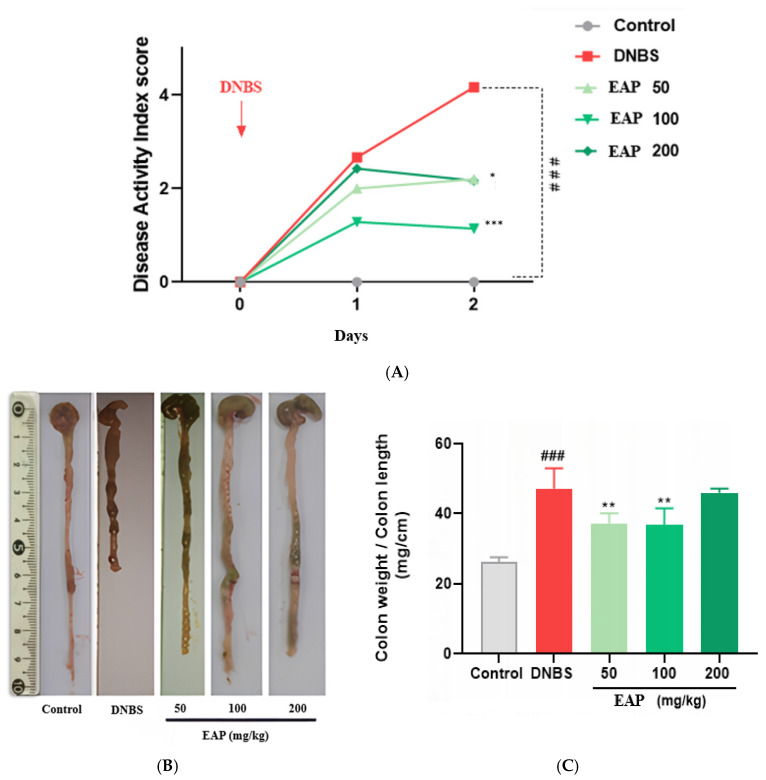
Effect of the ethanolic extract of *A. platensis* (EAP) on clinical and morphological parameters in DNBS-induced acute colitis. (**A**) Disease Activity Index (DAI). (**B**) Representative macroscopic appearance of colons. (**C**) Colon weight-to-length ratio. Mice were treated orally with EAP at doses of 50, 100, and 200 mg/kg. Data are expressed as mean ± SD (*n* = 7 per group). Statistical analysis was performed using one-way ANOVA followed by Dunnett’s multiple comparisons test. * *p* < 0.05, ** *p* < 0.01, *** *p* < 0.001 vs. the DNBS group; ### *p* < 0.001 vs. the healthy control group.

**Figure 3 ijms-26-05692-f003:**
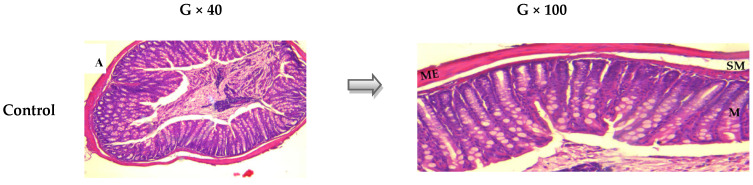
Histological evaluation of colonic tissues following treatment with the ethanolic extract of *A. platensis* (EAP) in DNBS-induced acute colitis in mice. Representative photomicrographs of hematoxylin–eosin-stained transverse colon sections (G × 40 and G × 100), illustrating (**A**) normal histology in the control group, (**B**) severe histopathological damage in the DNBS group, and (**C**–**E**) histological improvements in groups treated with EAP at doses of 50, 100, and 200 mg/kg, respectively. Key: (
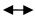
) submucosal edema; (
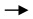
) infiltration of mononuclear cells in the submucosa; (
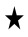
) disruption of crypt integrity; (
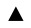
) necrotic areas. M: mucosa. SM: submucosa. ME: muscularis externa.

**Figure 4 ijms-26-05692-f004:**
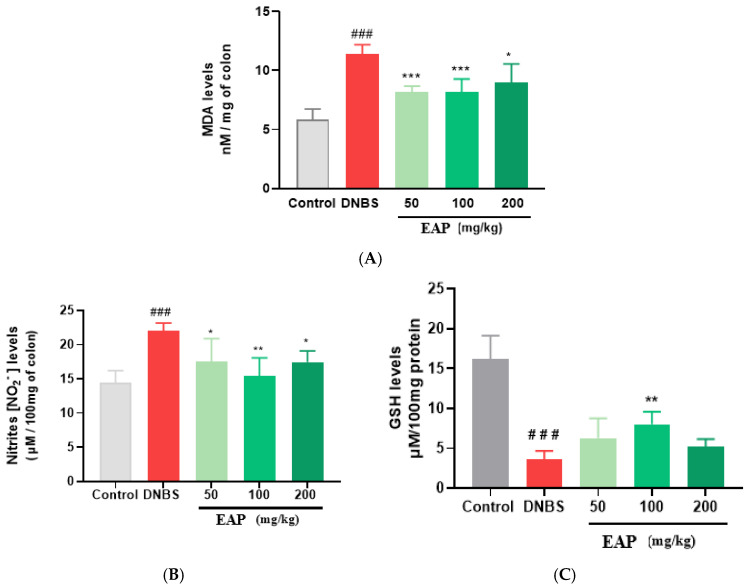
Effects of the ethanolic extract of *A. platensis* (EAP) on biochemical markers in DNBS-induced acute colitis in BALB/c mice. (**A**) Malondialdehyde (MDA) levels, (**B**) nitrite concentrations (NO), and (**C**) reduced glutathione (GSH) levels in colonic tissues following treatment with EAP at doses of 50, 100, and 200 mg/kg. Data are presented as mean ± standard deviation (SD), (*n* = 7 mice per group). Statistical analysis was performed using one-way ANOVA followed by Dunnett’s multiple comparisons test. * *p* < 0.05, ** *p* < 0.01, *** *p* < 0.001 vs. the DNBS group; ### *p* < 0.001 vs. the control group.

**Figure 5 ijms-26-05692-f005:**
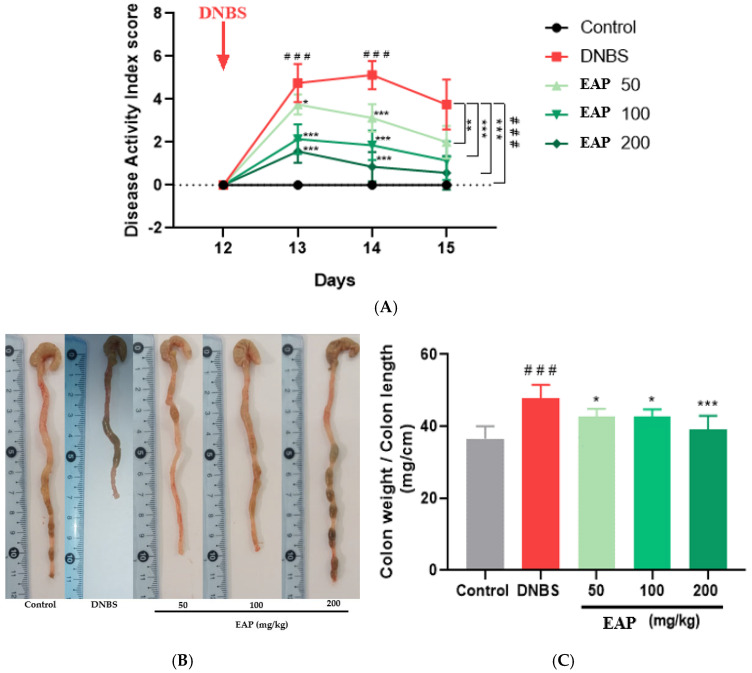
Effect of the ethanolic extract of *A. platensis* (EAP) on clinical and morphological parameters in DNBS-induced sub-chronic colitis in BALB/c mice. (**A**) Disease Activity Index (DAI). (**B**) Representative macroscopic appearance of the colon. (**C**) Colon weight-to-length ratio. Mice were treated orally with EAP at doses of 50, 100, and 200 mg/kg. Data are expressed as mean ± standard deviation (SD) (*n* = 7 per group). Statistical analysis was performed using one-way ANOVA followed by Dunnett’s multiple comparisons test. * *p* < 0.05, ** *p* < 0.01, *** *p* < 0.001 vs. the DNBS group; ### *p* < 0.001 vs. the control group.

**Figure 6 ijms-26-05692-f006:**
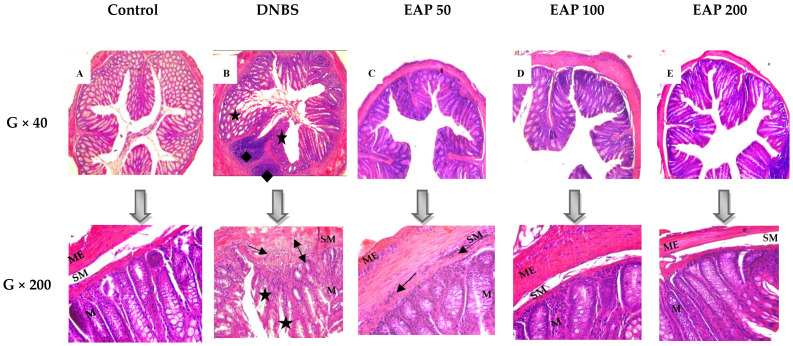
Histological evaluation of colonic tissue following treatment with the ethanolic extract of *A. platensis (EAP)* in DNBS-induced sub-chronic colitis in mice. Representative photomicrographs of hematoxylin–eosin-stained transverse colon sections, analyzed under light microscopy (G × 40 and G × 200), illustrate (**A**) normal histology in the control group, (**B**) severe histopathological damage in the DNBS group, and (**C**–**E**) histological improvements in groups treated with EAP at doses of 50, 100, and 200 mg/kg, respectively. Key: (
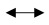
) submucosal edema; (
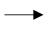
) infiltration of mononuclear cells in the submucosa; (
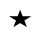
) disruption of crypt integrity; (
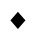
) granulomas. M: mucosa. SM: submucosa. ME: muscularis externa.

**Figure 7 ijms-26-05692-f007:**
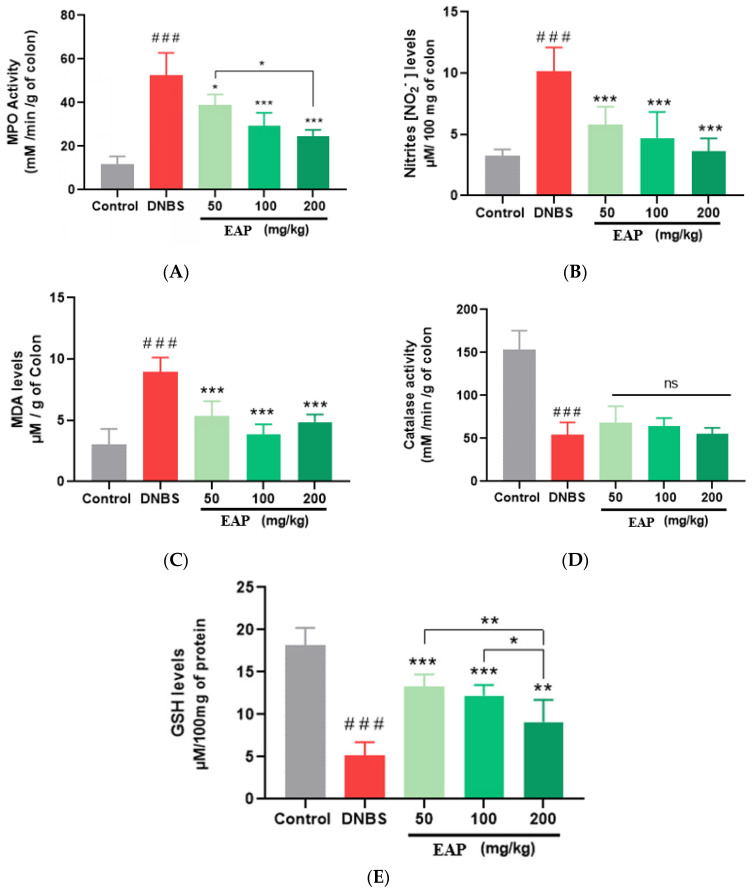
Effects of the ethanolic extract of *A. platensis* (EAP) on biochemical markers in DNBS-induced sub-chronic colitis in BALB/c mice. (**A**) Myeloperoxidase (MPO) activity; (**B**) nitrite (NO) levels; (**C**) malondialdehyde (MDA) content; (**D**) catalase activity; and (**E**) reduced glutathione (GSH) levels in colonic tissues following oral administration of EAP at doses of 50, 100, and 200 mg/kg. Data are expressed as mean ± standard deviation (SD). (*n* = 7 per group). Statistical analysis was performed using one-way ANOVA followed by Dunnett’s multiple comparisons test. ns, *p* > 0.05; * *p* < 0.05, ** *p* < 0.01, *** *p* < 0.001 vs. the DNBS group; ### *p* < 0.001 vs. the control group.

**Figure 8 ijms-26-05692-f008:**
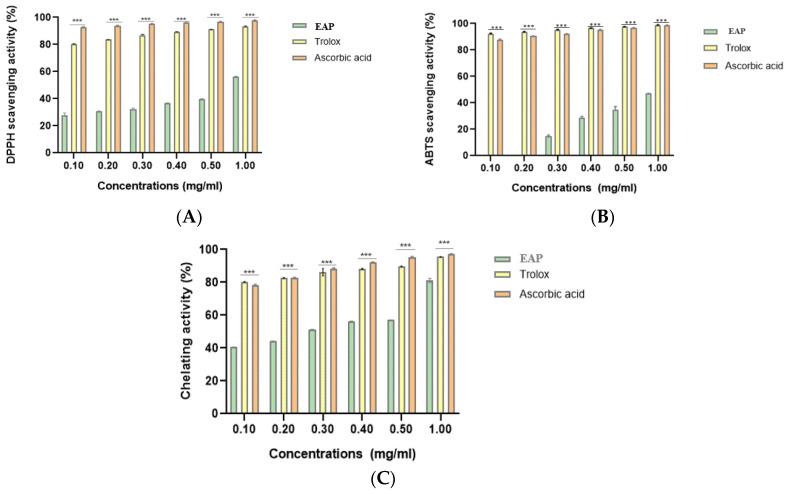
Antioxidant activity of the ethanolic extract of *A. platensis* (EAP) compared with Trolox and ascorbic acid. (**A**) DPPH• radical-scavenging activity; (**B**) ABTS•^+^ radical-scavenging activity; (**C**) ferrous ion-chelating (FIC) activity. Data are expressed as mean ± standard deviation (SD) (n = 3 independent replicates). Statistical analysis was conducted using one-way ANOVA followed by Tukey’s multiple comparisons test. *** *p* < 0.001 vs. Trolox/Ascorbic acid.

**Figure 9 ijms-26-05692-f009:**
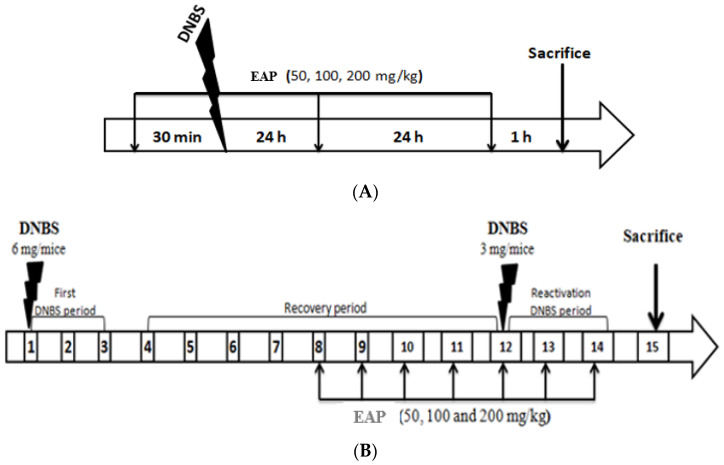
Schematic representation of the experimental protocols for DNBS-induced acute (**A**) and sub-chronic (**B**) colitis in BALB/c mice, including treatment timelines with the ethanolic extract of *A. platensis* (EAP).

**Table 1 ijms-26-05692-t001:** Identified phenolic compounds in the ethanolic extract of *A. platensis* as determined by HPLC-DAD-ESI-MS.

Phenolic Compounds		R_t_ (min)	λ_max_ (nm)	[M + H] (*m*/*z*)	Concentration (mg g^−1^ Extract)
1	Pyrogallol	2.92	270	127	3.142
2	3-Hydroxybenzoic acid	3.37	270	139	0.284
3	2,4 Dihydroxybenzoic acid	3.79	270	155	0.581
4	3,5 Dihydroxybenzoic acid	4.21	270	155	0.597
5	Chlorogenic acid	12.20	330	355	0.703
6	Vanilic acid	12.98	280	169	0.002
7	p-Coumaric acid	16.83	331	165	0.444
8	Ferulic acid	17.69	331	195	1.024
	Total phenolics				6.777

**Table 2 ijms-26-05692-t002:** IC_50_ values for the antioxidant activities of the ethanolic extract of *A. platensis* (EAP) and reference standards.

	IC_50_ (mg/mL)		
	EAP	Trolox	Ascorbic Acid
DPPH• scavenging activity	0.871 ± 0.012	0.081 ± 0.001	0.067 ± 0.00004
ABTS•^+^ scavenging activity	0.675 ± 0.026	0.068 ± 0.00009	0.072 ± 0.0001
Ferrous ion-chelating activity	0.213 ± 0.001	0.081 ± 0.001	0.083 ± 0.00003

**Table 3 ijms-26-05692-t003:** Antifungal activity of the ethanolic extract of *A. platensis* (EAP) against *C. albicans* and *A. flavus*.

Microorganisms	Inhibition Zone (mm)		
	EAP (mg.mL^−1^)		Ketoconazole (µg/disc)
	50	100	10
*Candida albicans*	15 ± 0.2	20 ± 0.1	30 ± 0.05
*Aspergillus flavus*	12 ± 0.3	15 ± 0.1	20 ± 0.05

## Data Availability

All data generated or analyzed during this study are included in this published article.

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
