# Peer review of "Phenolic Profiling and Bioactive Properties of Arthrospira platensis Extract in Alleviating Acute and Sub-Chronic Colitis"

_ijms, 2025, doi:10.3390/ijms26125692_

Round 1

Reviewer 1 Report

Comments and Suggestions for Authors

The manuscript, which investigates the potential of Arthrospira platensis (Spirulina) extract in mitigating acute and sub chronic colitis, presents novel insights into the use of plant-derived compounds for the management of inflammatory bowel disease. The topic is relevant, and the results could contribute to the growing body of evidence supporting functional foods in gastrointestinal health.

However, several improvements are necessary to enhance the scientific quality, clarity, and readability of the manuscript.

The current title, "Insights from Phenolic Profiling and Bioactivity Assays of Arthrospira platensis (Spirulina) Extract Alleviates Acute and Sub chronic Colitis", is overly long, grammatically incorrect, and structurally ambiguous. It combines a descriptive phrase with a conclusive verb, which leads to confusion.

Please revise the title to improve clarity and accuracy. 

The manuscript contains numerous grammatical and syntactical errors that hinder readability. A thorough English language revision is strongly recommended throughout the entire text.

Lines 28 and 43: The term “therapeutic” is used in an overly assertive way. Consider rephrasing these sentences to reflect potential or exploratory findings, rather than definitive therapeutic claims.

Quantitative results described in the text should consistently include measures of variability. Please include standard deviations alongside all reported mean values.

The names of the major phenolic compounds identified should be explicitly listed in the abstract to highlight the phytochemical relevance of the extract.

Figure 1: The quality of the figure needs improvement. Please enhance its resolution, clarity, and layout.
Figure 2C: Contains visual or formatting issues that require correction.
Figure 3: This figure also requires better formatting and possibly clearer annotation or labelling.
Consider presenting comparative results for acute and sub chronic colitis side by side where appropriate to facilitate interpretation, since the same methodologies were applied.

The three antioxidant activity assays performed are all chemical-based. To strengthen the discussion and support mechanistic insights, the inclusion of in vitro, cellular, antioxidant assays are recommended.

The rationale behind the anti fungal assays is unclear in the context of this work, testing colitis models chemically induced. Please clarify:

Is this disease model perhaps associated with Candida albicans or Aspergillus flavus?
How does anti fungal activity relate mechanistically to the alleviation of colitis in this study?
Please provide a more explicit justification for including this assay and connect it clearly to the study's hypothesis or broader implications.

Line 563–564: The sentence "Complementary, in vitro assays further confirmed the antioxidant capacity of the EAP extract, which were conducted in triplicate to ensure reproducibility and minimise experimental variability." is vague. Please specify which assays were performed. This clarification is essential for readers both within and outside the field.

The manuscript addresses a timely and interesting topic, but significant revisions are necessary to improve clarity, methodological coherence, and scientific rigour. The authors are encouraged to address the above points comprehensively to enhance the manuscript's overall quality and impact.

Author Response

Comment 1: The manuscript contains numerous grammatical and syntactical errors that hinder readability. A thorough English language revision is strongly recommended throughout the entire text.

Response 1: We appreciate the reviewer’s feedback regarding the language quality. In response, the entire manuscript has been thoroughly revised to improve grammar, syntax, and clarity. We believe the current version significantly enhances the readability and overall quality of the manuscript.

Comment 2: The current title, "Insights from Phenolic Profiling and Bioactivity Assays of Arthrospira platensis (Spirulina) Extract Alleviates Acute and Sub chronic Colitis", is overly long, grammatically incorrect, and structurally ambiguous. It combines a descriptive phrase with a conclusive verb, which leads to confusion. Please revise the title to improve clarity and accuracy.

Response 2: Thank you for this helpful observation. We agree that the original title was too long and grammatically inconsistent. In response, we have revised the title to improve clarity, accuracy, and structure. The new title is:

“Phenolic Profiling and Bioactive Properties of Arthrospira platensis Extract in Alleviating Acute and Sub-chronic Colitis.”

Comment 3: Lines 28 and 43: The term “therapeutic” is used in an overly assertive way. Consider rephrasing these sentences to reflect potential or exploratory findings, rather than definitive therapeutic claims.

Response 3: Thank you for your insightful comment. We fully agree that the use of the term “therapeutic” should be more cautiously framed given the preclinical nature of our study. Accordingly, we have rephrased both sentences to reflect the potential of Arthrospira platensis extract rather than making definitive claims. The revised sentences are as follows:

Line 28: “…making it a promising natural source for health-promoting applications.”

Line 43: “…support the potential of the EAP as a natural agent for managing intestinal inflammation and oxidative stress…”.

Comment 4: Quantitative results described in the text should consistently include measures of variability. Please include standard deviations alongside all reported mean values.

Response 4: We sincerely thank the reviewer for this valuable suggestion regarding the importance of reporting variability in quantitative data. In response, we would like to clarify that all experimental data derived from biological replicates or in vitro assays are indeed reported as mean ± standard deviation (SD) throughout the manuscript, as appropriate.

However, regarding the phenolic composition data, we would like to clarify that the analysis was performed only once on a single, well-characterized extract, and not under repeated or replicated conditions. As such, the reported values correspond to absolute concentrations obtained from a single analytical run. Therefore, standard deviations were not applicable in this specific case.

To address this point, we have added a clarification in the Methods section (3.3 section), explicitly stating that these specific values represent a single measurement.

Comment 5: The names of the major phenolic compounds identified should be explicitly listed in the abstract to highlight the phytochemical relevance of the extract.

Response 5: Thank you for your valuable suggestion. As recommended, we have revised the abstract to explicitly mention the major phenolic compounds identified in the ethanolic extract of Arthrospira platensis. Specifically, Pyrogallol, Ferulic acid, and Chlorogenic acid, which were the most abundant constituents, have now been clearly stated in the abstract to highlight the phytochemical relevance of the extract. The revised sentence reads:

“Phenolic profiling identified eight major compounds, with a cumulative content of 6.777 mg/g of extract, with Pyrogallol, Ferulic acid, and Chlorogenic acid were the most abundant.”

Comment 6:

Figure 1: The quality of the figure needs improvement. Please enhance its resolution, clarity, and layout.

Figure 2C: Contains visual or formatting issues that require correction.
Figure 3: This figure also requires better formatting and possibly clearer annotation or labelling.

Response 6: Thank you for your careful evaluation and helpful suggestions regarding the figures. In response:

  • Figure 1 has been replaced with a higher-resolution version to improve visual quality, clarity, and layout.
  • Figure 2C has been corrected to resolve the visual and formatting issues, ensuring accurate representation and consistent formatting with the rest of the panel.
  • Figure 3 has been reformatted for improved clarity, with enhanced annotations and labeling to aid reader comprehension.

Comment 7: Consider presenting comparative results for acute and sub-chronic colitis side by side where appropriate to facilitate interpretation, since the same methodologies were applied.

Response 7: We appreciate your suggestion regarding the comparative presentation of acute and sub-chronic colitis results. While we acknowledge that juxtaposing the data could help visualize differences and similarities more directly, we believe that in this case, a side-by-side format would have compromised clarity due to the complexity and density of the data.

Although similar methodologies were applied, key differences exist between the acute and sub-chronic colitis protocols:

  • The treatment was administered at three different doses, which already increases the volume of data for each model.
  • In the sub-chronic model, additional parameters were investigated, such as myeloperoxidase (MPO) and Catalase enzymatic activities, which were not assessed in the acute phase.
  • The negative and positive control groups differ between the two models and reflect the specific requirements of each inflammatory stage.

For these reasons, the authors have opted to present the data separately by inflammatory phase, which we believe allows for greater clarity, avoids visual overload, and respects the methodological distinctions between the two models.

Comment 8: The three antioxidant activity assays performed are all chemical-based. To strengthen the discussion and support mechanistic insights, the inclusion of in vitro, cellular, antioxidant assays are recommended.

Response 8: We thank the reviewer for this insightful and constructive suggestion. We fully agree that the inclusion of cellular antioxidant assays would provide deeper mechanistic insights into the antioxidant potential of Arthrospira platensis extract and enhance the biological relevance of the findings. However, the present study was designed as an initial comprehensive screening, employing widely accepted and standardized chemical-based assays (DPPH, ABTS, and ferrous ion-chelating activity), which offer a robust and reproducible assessment of the extract’s radical scavenging and metal-chelating capacities, particularly suitable for complex natural matrices.

We acknowledge the value of complementing these results with cell-based assays, such as intracellular ROS quantification, antioxidant enzyme induction, or ARE-luciferase reporter assays. This important aspect has been clearly stated in the revised manuscript (lines 577–583, Discussion) as a limitation, and we have proposed the inclusion of such approaches in future investigations to better elucidate the extract’s mode of action at the cellular level, especially within relevant intestinal epithelial models.

Comment 9: The rationale behind the anti-fungal assays is unclear in the context of this work, testing colitis models chemically induced. Please clarify:

Is this disease model perhaps associated with Candida albicans or Aspergillus flavus?
How does anti-fungal activity relate mechanistically to the alleviation of colitis in this study?
Please provide a more explicit justification for including this assay and connect it clearly to the study's hypothesis or broader implications.

Response 9: We appreciate the reviewer’s comment regarding the inclusion of antifungal assays. Although the DNBS-induced colitis model is chemically triggered, it recapitulates key features of Crohn’s disease, including mucosal damage, oxidative stress, and immune dysregulation. Notably, recent evidence highlights the role of intestinal fungal dysbiosis, particularly the overgrowth of Candida albicans, in the exacerbation of inflammatory bowel diseases. Fungi such as C. albicans can impair epithelial integrity and modulate host immune responses, contributing to sustained intestinal inflammation. Therefore, assessing the antifungal potential of the ethanolic extract of Arthrospira platensis (EAP) against C. albicans offers additional insight into its capacity to counteract pathobionts that may aggravate colitis, thus broadening the scope of its therapeutic relevance.

Furthermore, the inclusion of Aspergillus flavus in the antifungal screening aimed to explore the broader antifungal spectrum of EAP, given its rich phytochemical profile. The antifungal assays were intended as a complementary approach to evaluate the multi-target bioactivity of EAP, particularly considering its potential use as a natural therapeutic agent in gastrointestinal disorders characterized by both inflammation and microbial imbalance. This rationale has been clarified in the discussion of the revised manuscript to ensure alignment between the in vitroantifungal data and the overall objectives of the study.

This clarification has been added to the Discussion section of the revised manuscript (with tracking changes, in red) to ensure proper alignment between the antifungal data and the overall study objectives, as well as to explicitly acknowledge the rationale of this approach.

Comment 10: Line 563–564: The sentence "Complementary, in vitro assays further confirmed the antioxidant capacity of the EAP extract, which were conducted in triplicate to ensure reproducibility and minimise experimental variability." is vague. Please specify which assays were performed. This clarification is essential for readers both within and outside the field.

Response 10: We thank the reviewer for this important observation. To clarify, the complementary in vitro antioxidant assays performed to confirm the capacity of the EAP extract included DPPH radical scavenging, ABTS radical cation decolorization, and ferrous ion-chelating activity assays. All tests were conducted in triplicate to ensure reproducibility and minimize experimental variability. This specification has been added in the revised manuscript for improved clarity and accessibility to readers from diverse scientific backgrounds.

Reviewer 2 Report

Comments and Suggestions for Authors

Title: Insights from Phenolic Profiling and Bioactivity Assays of Arthrospira platensis (Spirulina) Extract Alleviates Acute and Sub-chronic Colitis.

The title of the manuscript is consistent with the topic of the study. The Authors of this manuscript explored the profile of the phenolic constituents of A. platensis (EAP), the ethanolic extract using HPLC-DAD-ESI-MS, and evaluated its pharmacological effects in acute and sub-chronic experimental colitis, alongside its antioxidant and antifungal properties. Phenolic profiling identified eight major compounds, with a cumulative content of 6.777 mg/g extract. In vivo administration of EAP significantly reduced the Disease Activity Index (DAI), ameliorated macroscopic colonic damage, and preserved mucosal architecture in both inflammatory phases. Biochemical analyses revealed significant reductions in myeloperoxidase (MPO) activity, nitric oxide (NO), and malondialdehyde (MDA) levels, accompanied by increased glutathione (GSH) content and catalase activity. In vitro, EAP demonstrated 56% DPPH and 47% ABTS radical scavenging activity, 81% ferrous ion-chelating capacity, and antifungal activity with inhibition zones of 20 mm and 15 mm against Candida albicans and Aspergillus flavus, respectively. These findings highlight the multi-target bioactivity of EAP and support its potential as a natural therapeutic agent for mitigating intestinal inflammation and oxidative stress across both acute and sub-chronic phases.

The work contains factual and methodological errors, especially in the section on research on the phytochemical profile of the tested extract. In the section of the discussion, the Authors draw constructive conclusions. The scope of literature data is up-to-date and consistent with the subject of the research undertaken.

Comments and suggestions for Authors:

  • Table 1. lacks fragmentation data (MS2 or MS3) for the interpreted compounds.
  • Why these compounds (Figure 1.) were analyzed in ESI positive. From my knowledge, these phenols ionize much better in ESI negative. Explain this, please.
  • The chromatographic separation of the first four compounds in Figure 1. is very poor. Couldn't the method be modified to make these separations better?
  • It is interesting that in Table 3., in the last column, the standard deviations are 0.0. The lack of a standard deviation may indicate that there were no repetitions in the study. Please explain this.
  • The situation with zero standard deviations is also in Table 2. in IC50 values. Please explain this.

In my opinion, the manuscript can only be published after major corrections have been made.

Author Response

Comment 1: Table 1. lacks fragmentation data (MS2 or MS3) for the interpreted compounds.

Thank you for your valuable comment regarding the inclusion of fragmentation data (MS2 or MS3) for the identified phenolic compounds. We acknowledge the importance of detailed fragmentation data for compound identification. In our study, as described in the section "3.3 Phenolic Profiling of EAP Extract Using HPLC-DAD-ESI-MS," we used an HPLC-DAD-ESI-MS system equipped with a single quadrupole MS detector. As such, our instrumentation did not allow for MS2 or MS3 fragmentation but provided reliable [M+H]+ (m/z) ions, which are presented in Table 1 (column [4]). We have noted this methodological limitation in the revised manuscript and provided an explicit statement regarding the ionization mode and instrumentation in the Materials and Methods section. If the opportunity arises in future studies, we plan to include MS2/MS3 data to further strengthen our compound identification protocol.

Comment 2: Why these compounds (Figure 1.) were analyzed in ESI positive. From my knowledge, these phenols ionize much better in ESI negative. Explain this, please.

We sincerely appreciate the reviewer’s insightful observation regarding the ionization behavior of phenolic compounds in ESI modes. It is well established that many phenolic acids, such as hydroxybenzoic and hydroxycinnamic acids (e.g., 3-hydroxybenzoic acid, 2,4-dihydroxybenzoic acid, chlorogenic acid, p-coumaric acid, and ferulic acid), exhibit better ionization efficiency in the negative ion mode due to their acidic functional groups (carboxylic acid moieties). This is a known phenomenon and is typically preferred when focusing solely on phenolic profiling.

  • Our target analytes were not limited to phenolic acids but also included other classes of metabolites, such as basic and neutral compounds, which ionize more efficiently in positive mode. For instance, pyrogallol, despite being a polyphenol, is a neutral molecule (pKa ~9.1) and shows appreciable ionization under ESI positive conditions. Running the analysis in positive mode enabled us to capture these co-existing metabolites in a single acquisition, avoiding potential loss of information.
  • Our HPLC-MS system was pre-optimized for ESI positive mode, with parameters (such as capillary voltage, desolvation gas flow, and cone voltage) fine-tuned to maximize sensitivity and signal-to-noise ratio in this polarity. This setup allowed for robust and reproducible detection of the [M+H]+ ions, even for acidic compounds with inherently lower ionization efficiency in positive mode. The observed signal intensities, although slightly lower than what might be obtained in negative mode, were sufficient for confident detection and tentative identification based on accurate mass and fragmentation patterns.
  • While negative mode enhances the ionization of acidic phenolics, it can suppress signals from non-acidic compounds or lead to ion suppression effects due to matrix complexity. By operating in positive mode, we minimized such limitations and ensured the detection of a wider chemical space, including neutral and basic compounds that might co-elute in the chromatographic separation.
  • We fully acknowledge that complementary analysis in negative mode would further improve the coverage of phenolic acids and enhance quantification accuracy. We plan to integrate negative ESI mode in future studies to provide a more comprehensive metabolomic fingerprint of the extract, particularly focusing on the differential ionization behavior of phenolic subclasses.

In summary, while ESI negative mode is indeed optimal for many phenolic acids, the positive mode was chosen here to enable a broader screening of co-existing metabolites, maintain consistency with instrumental parameters, and achieve sufficient sensitivity for compound detection. We appreciate the reviewer’s suggestion and will consider negative mode analysis in future experiments for a more complete characterization.

Comment 3: The chromatographic separation of the first four compounds in Figure 1. is very poor. Couldn't the method be modified to make these separations better?

Thank you for your valuable observation. We acknowledge that the chromatographic resolution of the first four peaks in Figure 1 appears limited. These compounds, highly polar phenolic acids such as pyrogallol, 3-hydroxybenzoic acid, and dihydroxybenzoic acid derivatives, are known to elute closely in reversed-phase chromatography due to their similar polarity and molecular weight.

In our study, chromatographic separation was performed using a Kinetex XB-C18 column and a carefully optimized multistep gradient with 0.1% formic acid in water and acetonitrile as mobile phases. The elution gradient was designed to accommodate a wide range of phenolic compounds, from highly polar to moderately lipophilic, within a single run of 30 minutes, ensuring detection of both early- and late-eluting constituents.

While a longer or shallower gradient at the initial phase might improve the separation of the early-eluting compounds, this would inevitably prolong the analysis time and risk loss of resolution or peak shape integrity for mid- and late-eluting phenolics (e.g., chlorogenic acid, ferulic acid).

Importantly, although DAD chromatograms at λ = 280 nm show a visual overlap, all compounds were confidently identified and distinguished using combined data from retention time, UV spectra, and accurate m/z values obtained via ESI-MS in positive ionization mode. This multi-dimensional identification ensures specificity despite partial co-elution.

That said, we fully recognize this limitation and appreciate the suggestion. In future work, improved separation of these early peaks could be explored through alternative stationary phases (e.g., polar-embedded or phenyl-hexyl columns), column dimensions, or targeted gradient optimization, especially if early phenolics are of primary focus.

Comment 4: It is interesting that in Table 3., in the last column, the standard deviations are 0.0. The lack of a standard deviation may indicate that there were no repetitions in the study. Please explain this.

The situation with zero standard deviations is also in Table 2. in IC50 values. Please explain this.

Thank you for your observation. We would like to clarify that standard deviations were indeed calculated based on triplicate measurements. However, the SD values were very small (approximately ± 0.00004), and due to rounding to two decimal places, they appear as 0.00 in the table. This does not reflect a lack of replicates but rather the high reproducibility of the data. For transparency, we provide below the same table with the exact standard deviation values included.

IC50 (mg/ml)

APH extract

Trolox

Ascorbic acid

DPPH- scavenging activity

0,871 ± 0,012

0,081 ± 0,001

0,067 ± 0,00004

ABTS-scavenging activity

0,675 ± 0,026

0,068 ± 0,00009

0,072 ± 0,0001

Ferrous ion-chelating activity

0,213 ± 0,001

0,081 ± 0,001

0,083 ± 0,00003

Reviewer 3 Report

Comments and Suggestions for Authors

The manuscript is well presented and the data is interesting. However, I recommend some corrections before publication. More specifically:

Lines 57-59: The authors mention “Epidemiological studies indicate considerable geographic and environmental variability, with incidence highest in industrialized nations and increasing in newly industrialized regions.” but no studies are cited. When the literature is mentioned, then there should also be the corresponding studies to which the authors refer.

Line 74: When the Latin name is mentioned for the first time, it should be given fully written i.e. Arthrospira platensis, and then it should be written as A. platensis. Also, all Latin names should be written in italics. Please revise in the whole manuscript.

Table 1: Please correct Pyrogallol

Author Response

Comment 1: Lines 57-59: The authors mention “Epidemiological studies indicate considerable geographic and environmental variability, with incidence highest in industrialized nations and increasing in newly industrialized regions.” but no studies are cited. When the literature is mentioned, then there should also be the corresponding studies to which the authors refer.

Thank you for your observation. We agree that statements referring to literature should be supported by appropriate citations. In response, we have now cited recent and relevant studies [Lin et al., 2024; Dou et al., 2024] to substantiate the statement on geographic and environmental variability in IBD incidence.

Comment 2: Line 74: When the Latin name is mentioned for the first time, it should be given fully written i.e. Arthrospira platensis, and then it should be written as A. platensis. Also, all Latin names should be written in italics. Please revise in the whole manuscript.

Thank you for your comment. We have corrected the text to ensure that the full Latin name Arthrospira platensis is written out at its first mention and abbreviated as A. platensis thereafter. All Latin names throughout the manuscript have also been revised to appear in italics, as per scientific writing conventions.

Comment 3: Table 1: Please correct Pyrogallol

Thank you for pointing this out. The spelling error has been corrected in Table 1, and Pyrogallol is now properly written, as indicated in red in the revised manuscript.

Reviewer 4 Report

Comments and Suggestions for Authors

The manuscript, "Insights from Phenolic Profiling and Bioactivity Assays of Arthrospira platensis (Spirulina) Extract Alleviates Acute and Sub-chronic Colitis", contains interesting data, but a few points need clarifying.
1) What does Figure 1 represent? Is it a DAD or an MS chromatogram? The figure 1 description is incorrect – part of it should be included in the Materials and Methods section. When describing chromatograms, information about the chromatographic system (e.g. column type, gradient profile, temperature, flow rate) is usually provided.
2) Tab 1. Are the Authors sure that the maximum absorption for p-coumaric acid is 331 nm? From the reviewer's experience, the maximum absorption for this acid is 310 nm.
3) Section 3.3, lines 351–352: Are the Authors referring to the range of data recording? When using a DAD detector, spectra should be recorded in the range of about 190 nm to 400 nm for phenolic acids.
Are the Authors sure that the phenolic acids are responsible for the observed activity of the extract? Given the extract's composition, can therapeutic concentrations be achieved in vivo? Did the Authors consider analysing the extract to identify and determine metabolites from other chemical groups?
5) The Authors should discuss the antifungal activity in more detail, especially in light of the positive control results.
6) The Authors should present the DAD and MS spectra of the identified compounds in the supplementary materials.

Author Response

Comment 1:  What does Figure 1 represent? Is it a DAD or an MS chromatogram? The figure 1 description is incorrect – part of it should be included in the Materials and Methods section. When describing chromatograms, information about the chromatographic system (e.g. column type, gradient profile, temperature, flow rate) is usually provided.

Response 1: Thank you for your valuable comment. Figure 1 represents the HPLC-DAD chromatogram of phenolic compounds detected in the ethanolic extract of Arthrospira platensis, recorded at 280 nm. We acknowledge that the original figure legend did not clearly indicate this distinction. To address this, we have revised the figure legend to explicitly state that it corresponds to the DAD chromatogram. Furthermore, all relevant experimental details, including the chromatographic system, column type, gradient conditions, flow rate, detection wavelengths, and MS acquisition parameters, are already provided in the Materials and Methods section to ensure full transparency and reproducibility. The revised figure legend now reads:

Figure 1. HPLC-DAD chromatogram of phenolic compounds identified in the ethanolic extract of A. platensis.

Comment 2: Tab 1. Are the Authors sure that the maximum absorption for p-coumaric acid is 331 nm? From the reviewer's experience, the maximum absorption for this acid is 310 nm.

Response 2:  We thank the reviewer for raising this important point regarding the maximum absorption wavelength (λmax) of p-coumaric acid. We acknowledge that the reported λmax for p-coumaric acid can vary depending on the analytical conditions. While it is often cited near 310 nm in neutral solvents such as methanol or water, numerous studies have demonstrated that the λmax of p-coumaric acid shifts toward higher wavelengths under acidic conditions. This is due to changes in the protonation state of the phenolic moiety, which affects its electronic transitions. For example, studies such as (Anastasiadi et al., 2010) and (Paniagua-García et al., 2019) reported λmax values around 320 nm for p-coumaric acid in mobile phases containing acetic acid or other acidic modifiers. These findings align with well-established principles of UV-Vis spectroscopy, where solvent polarity and pH can cause bathochromic shifts in the absorption spectra of phenolic compounds.

In our study, the phenolic profiling was conducted using HPLC-DAD-ESI-MS with an acidic mobile phase consisting of 0.1% formic acid in water and 0.1% formic acid in acetonitrile. Chromatograms were recorded at 280 nm and 340 nm, and the λmax values of individual phenolic compounds were directly obtained from the diode-array detector (DAD). Under these specific acidic chromatographic conditions, we observed that p-coumaric acid eluted at a retention time of 16.83 minutes and exhibited a λmax at 331 nm. This result is consistent with the bathochromic shifts reported in similar studies using acidic solvents, further confirming the influence of experimental conditions on the optical properties of phenolic compounds.

In summary, we are confident that the λmax of 331 nm for p-coumaric acid in our study is accurate and justified, given the acidic nature of the mobile phase employed.

  • References

Anastasiadi, M., Pratsinis, H., Kletsas, D., Skaltsounis, A.-L., Haroutounian, S.A., 2010. Bioactive non-coloured polyphenols content of grapes, wines and vinification by-products: Evaluation of the antioxidant activities of their extracts. Food Res. Int. 43, 805–813. https://doi.org/10.1016/j.foodres.2009.11.017

Paniagua-García, A.I., Hijosa-Valsero, M., Garita-Cambronero, J., Coca, M., Díez-Antolínez, R., 2019. Development and validation of a HPLC-DAD method for simultaneous determination of main potential ABE fermentation inhibitors identified in agro-food waste hydrolysates. Microchem. J. 150, 104147. https://doi.org/10.1016/j.microc.2019.104147

Comment 3: Section 3.3, lines 351–352: Are the Authors referring to the range of data recording? When using a DAD detector, spectra should be recorded in the range of about 190 nm to 400 nm for phenolic acids.

Response 3: Thank you for this comment. We would like to clarify that full UV-Vis spectral data were acquired for all peaks over the range of 190 to 400 nm, in accordance with established protocols for phenolic acid analysis by HPLC-DAD. This comprehensive spectral acquisition allows precise determination of maximum absorption wavelengths (λmax) necessary for compound identification.

The wavelengths 280 nm and 340 nm mentioned in the manuscript correspond exclusively to the chromatographic detection channels used for quantification, not to the spectral acquisition range. These wavelengths were chosen to optimize sensitivity for different classes of phenolic compounds but do not limit the spectral scan range.

Therefore, we confirm that the spectral data acquisition fully complies with standard practices, ensuring accurate qualitative and quantitative analysis of phenolic acids in our samples.

Comment 4: Are the Authors sure that the phenolic acids are responsible for the observed activity of the extract? Given the extract's composition, can therapeutic concentrations be achieved in vivo? Did the Authors consider analysing the extract to identify and determine metabolites from other chemical groups?

Response 4: We thank the reviewer for this valuable comment. While we identified several phenolic acids in our extract, such as p-coumaric acid, ferulic acid, chlorogenic acid, and pyrogallol, well known for their antioxidant, anti-inflammatory, and antimicrobial activities, we fully acknowledge that these compounds alone cannot account for the full spectrum of biological effects observed. As detailed in our discussion (lines 546–558), the bioactivity of these phenolic acids is supported by mechanistic evidence: pyrogallol modulates the NF-κB pathway, reducing pro-inflammatory cytokines; chlorogenic acid and ferulic acid activate Nrf2/ARE signaling, enhancing antioxidant defenses and suppressing lipid peroxidation; all contribute to downregulating COX-2 and iNOS pathways, mitigating both oxidative and nitrosative stress.

However, we also recognize that A. platensis is a complex matrix containing multiple classes of bioactive compounds, including phycobiliproteins (such as phycocyanin), polysaccharides, fatty acids, and amino acids. These molecules may exert individual biological effects and, more importantly, act synergistically with phenolic acids to enhance the overall activity of the extract. Therefore, while phenolic acids are major contributors, we do not attribute the observed bioactivity solely to them.

Regarding the relevance of the concentrations used, the doses selected for this study were carefully chosen based on existing literature and preliminary data, aiming to ensure pharmacologically relevant levels that could be realistically achieved in vivo while remaining within a safe, non-toxic range. Nonetheless, we acknowledge that further pharmacokinetic and bioavailability studies are essential to confirm the capacity of these compounds to reach effective concentrations in target tissues and to fully validate the therapeutic potential of the extract in vivo.

Finally, while our study focused on phenolic acids due to their well-established roles in modulating inflammation and oxidative stress, we agree that a broader metabolite profiling is necessary. We plan to extend our chemical analysis to include other bioactive compounds, which will allow for a more comprehensive understanding of the extract’s therapeutic potential.

In summary, the biological effects of A. platensis likely result from the interplay of multiple bioactive compounds, with phenolic acids as important, but not exclusive, contributors. Further work is necessary to elucidate the complete pharmacological profile and confirm the in vivo relevance of these findings

Comment 5: The Authors should discuss the antifungal activity in more detail, especially in light of the positive control results.

Response 5: We thank the reviewer for this valuable comment. We have carefully revised the discussion to provide a more detailed and nuanced interpretation of the antifungal activity, particularly in comparison with the positive control (ketoconazole) as indicated in red in the revised manuscript.

Comment 6: The Authors should present the DAD and MS spectra of the identified compounds in the supplementary materials.

We appreciate the reviewer’s suggestion to include both the Diode Array Detector (DAD) and MS spectra of the identified compounds in the supplementary materials. In Table 1 of the revised manuscript, we have included DAD UV λmax (nm) values (column [5]) and MS data ([M+H]+ (m/z), column [4]) for all identified compounds.

Despite the instrumental limitations, the use of HPLC coupled with DAD and ESI-MS provided a robust and reliable platform for the characterization of phenolic compounds within the extract. The method employed in this research are internal validated and standardized in our lab, allowing for simultaneous collection of chromatographic, UV-Visible, and mass spectral data, enabling effective compound profiling and tentative identification. The DAD facilitated detection of characteristic UV absorbance maxima, supporting the classification of phenolic subclasses, while the ESI-MS provided key molecular ion information. The combination of these detection techniques enhanced confidence in compound identification and enabled comprehensive profiling, even in complex matrices.

We have revised the manuscript accordingly, and we have integrated this clarification into the Methods section (Section 3.3) to acknowledge the limitations of the method and to improve the transparency of the analytical approach.

Round 2

Reviewer 1 Report

Comments and Suggestions for Authors

The manuscript has been revised accordingly.

However, Figure 1 still requires adjustment to ensure consistency between the illustration and the legend text.

Author Response

Comment: The manuscript has been revised accordingly. However, Figure 1 still requires adjustment to ensure consistency between the illustration and the legend text.

Response: We are grateful for the reviewer's helpful observation. We acknowledge that while the overall analysis was HPLC-DAD-ESI-MS, Figure 1 exclusively displays the DAD chromatogram, which could create confusion. The mass spectrometry results are detailed separately in Table 1.

To ensure the figure caption is perfectly aligned with its content and to avoid any ambiguity, we have revised it as follows:

“DAD chromatogram of the ethanolic extract of Arthrospira platensis obtained by HPLC-DAD-ESI-MS analysis

(280 nm)

This new caption now accurately reflects the specific data presented in the figure.

Reviewer 2 Report

Comments and Suggestions for Authors

Title: Insights from Phenolic Profiling and Bioactivity Assays of Arthrospira platensis (Spirulina) Extract Alleviates Acute and Sub-chronic Colitis.

The title of the manuscript is consistent with the topic of the study. The Authors of this manuscript explored the profile of the phenolic constituents of A. platensis (EAP), the ethanolic extract using HPLC-DAD-ESI-MS, and evaluated its pharmacological effects in acute and sub-chronic experimental colitis, alongside its antioxidant and antifungal properties. Phenolic profiling identified eight major compounds, with a cumulative content of 6.777 mg/g extract. In vivo, administration of EAP significantly reduced the Disease Activity Index (DAI), ameliorated macroscopic colonic damage, and preserved mucosal architecture in both inflammatory phases. Biochemical analyses revealed significant reductions in myeloperoxidase (MPO) activity, nitric oxide (NO), and malondialdehyde (MDA) levels, accompanied by increased glutathione (GSH) content and catalase activity. In vitro, EAP demonstrated 56% DPPH and 47% ABTS radical scavenging activity, 81% ferrous ion-chelating capacity, and antifungal activity with inhibition zones of 20 mm and 15 mm against Candida albicans and Aspergillus flavus, respectively. These findings highlight the multi-target bioactivity of EAP and support its potential as a natural therapeutic agent for mitigating intestinal inflammation and oxidative stress across both acute and sub-chronic phases.

The Authors answered all my questions exhaustively.

Comments and suggestions for Authors:

  • Please include Tables 2 and 3 with full standard deviations.

In my opinion, the manuscript can only be published after minor corrections have been made.

Author Response

Comment: Please include Tables 2 and 3 with full standard deviations.

Response: We thank the reviewer for this important comment regarding data presentation. You are correct that rounding the standard deviation (SD) values to two decimal places masked the true, albeit small, variability for some of our data.

As requested, we have revised Tables 2 and 3 to present the full standard deviations with greater decimal precision. This ensures that the variability from our triplicate independent measurements is accurately and transparently represented. These changes have been implemented in the revised manuscript.

Reviewer 4 Report

Comments and Suggestions for Authors

The Authors clarified any uncertainties I had and made the necessary corrections to the manuscript. I believe it is now ready for publication. However, I think it would be good practice to present the DAD and MS spectra in the supplementary material.

Author Response

Comment: The Authors clarified any uncertainties I had and made the necessary corrections to the manuscript. I believe it is now ready for publication. However, I think it would be good practice to present the DAD and MS spectra in the supplementary material.

Response: We sincerely thank the reviewer for their positive assessment and this excellent suggestion.

We agree that including the DAD and MS spectra will enhance the manuscript's transparency and value. We are preparing this supplementary material now.

However, due to the upcoming national holidays (Rusalii/Whit Monday), we kindly request a brief extension for this addition. We will upload the supplementary file containing all relevant spectra no later than June 12, 2025.

We appreciate your understanding and your guidance in improving the final version of our paper.
